# Antibody Responses after a Third Dose of COVID-19 Vaccine in Kidney Transplant Recipients and Patients Treated for Chronic Lymphocytic Leukemia

**DOI:** 10.3390/vaccines9101055

**Published:** 2021-09-23

**Authors:** Julien Marlet, Philippe Gatault, Zoha Maakaroun, Hélène Longuet, Karl Stefic, Lynda Handala, Sébastien Eymieux, Emmanuel Gyan, Caroline Dartigeas, Catherine Gaudy-Graffin

**Affiliations:** 1INSERM U1259, Université de Tours et CHRU de Tours, 37000 Tours, France; karl.stefic@univ-tours.fr (K.S.); L.HANDALA@chu-tours.fr (L.H.); sebastien.eymieux@univ-tours.fr (S.E.); catherine.gaudy-graffin@univ-tours.fr (C.G.-G.); 2Service de Bactériologie-Virologie-Hygiène, CHRU de Tours, 37000 Tours, France; 3Transplantation rénale–Immunologie clinique, CHRU de Tours, 37000 Tours, France; philippe.gatault@univ-tours.fr (P.G.); H.LONGUET@chu-tours.fr (H.L.); 4Centre de vaccination, CHRU de Tours, 37000 Tours, France; Z.MAAKAROUN-VERMESSE@chu-tours.fr; 5Service de médecine pédiatrique, CHRU de Tours, 37000 Tours, France; 6Plate-Forme IBiSA de Microscopie Electronique, Université de Tours and CHRU de Tours, 37000 Tours, France; 7Hématologie et Thérapie Cellulaire, CHRU de Tours, 37000 Tours, France; E.GYAN@chu-tours.fr (E.G.); c.dartigeas@chu-tours.fr (C.D.)

**Keywords:** COVID-19, vaccine, kidney transplant, chronic lymphocytic leukemia, antibody

## Abstract

The impact of a third dose of COVID-19 vaccine on antibody responses is unclear in immunocompromised patients. The objective of this retrospective study was to characterize antibody responses induced by a third dose of mRNA COVID-19 vaccine in 160 kidney transplant recipients and 20 patients treated for chronic lymphocytic leukemia (CLL). Prevalence of anti-spike IgG ≥ 7.1 and ≥ 30 BAU/mL after the third dose were 47% (75/160) and 39% (63/160) in kidney transplant recipients, and 57% (29/51) and 50% (10/20) in patients treated for CLL. Longitudinal follow-up identified a moderate increase in SARS-CoV-2 anti-spike IgG levels after a third dose of vaccine in kidney transplant recipients (0.19 vs. 5.28 BAU/mL, *p* = 0.03) and in patients treated for CLL (0.63 vs. 10.7 BAU/mL, *p* = 0.0002). This increase in IgG levels had a limited impact on prevalence of anti-spike IgG ≥ 30 BAU/mL in kidney transplant recipients (17%, 2/12 vs. 33%, 4/12, *p* = 0.64) and in patients treated for CLL (5%, 1/20 vs. 45%, 9/20, *p* = 0.008). These results highlight the need for vaccination of the general population and the importance of non-medical preventive measures to protect immunocompromised patients.

## 1. Introduction

Vaccine effectiveness against symptomatic COVID-19 was estimated at between 70 and 97% after two doses of COVID-19 vaccine in immunocompetent patients [1,2,3]. In contrast, few immunocompromised patients have been enrolled in COVID-19 phase 3 clinical trials and vaccine effectiveness in this population remains unclear. In a recent observational study, solid organ transplant recipients had an 82-fold higher risk of breakthrough infection and a 485-fold higher risk of breakthrough infection with associated hospitalization and death [4]. Antibody responses after two doses of COVID-19 vaccine, defined as anti-spike IgG seroconversion, are impaired in these patients [5]. They were estimated between 38 to 42% for kidney transplant patients [5,6] and between 23 to 52% for patients with chronic lymphocytic leukemia, despite the fact that most of these CLL patients were not undergoing cancer therapy [7,8,9]. The impact of a third dose of COVID-19 vaccine on antibody responses is unclear in these patients. Response rates to a third dose ranged between 47 and 68% in kidney transplant recipients [10,11,12] and limited data are available in patients treated for chronic lymphocytic leukemia (CLL). The objective of this retrospective study was to characterize antibody responses induced by a third dose of mRNA COVID-19 vaccine in 160 kidney transplant recipients and 20 patients treated for CLL.

## 2. Materials and Methods

SARS-CoV-2 anti-spike IgG was tested for all patients at least 21 days after the second and/or third dose of mRNA COVID-19 vaccine, using a SARS-CoV-2 IgG II Quant assay on an Alinity i system (Abbott). Assay results in AU/mL were converted into BAU/mL (international standard units [13]) using a conversion factor of 0.142, according to the manufacturer’s recommendations. We defined responders as patients with positive SARS-CoV-2 IgG, corresponding to levels ≥ 7.1 binding antibody units (BAU)/mL, according to the manufacturer’s recommendations and previous studies [14,15,16,17,18]. This interpretation is limited by its lack of association with vaccine effectiveness and its lack of comparability with other assays, using different cut-offs. As such, results were also compared with a cut-off of 30 BAU/mL, recently associated with 50% vaccine effectiveness against symptomatic COVID-19 in immunocompetent patients [19]. All calculations were performed with GraphPad Prism 9.0 using Wilcoxon test (paired values), Mann-Whitney (unpaired values) and Fisher’s exact test (binomial variables). *p*-values < 0.05 (2-sided) were considered significant.

## 3. Results

### 3.1. Kidney Transplant Recipients

Prevalence of positive SARS-CoV-2 anti-spike IgG (≥7.1 BAU/mL) in kidney transplant recipients tested either after the second (n = 97) or third dose (n = 160) of COVID-19 vaccine was not significantly different (43%, 42/97 vs. 47%, 75/160, *p* = 0.61, Table 1). Prevalence of SARS-CoV-2 anti-spike IgG ≥ 30 BAU/mL was also not significantly different between these two groups (30%, 29/97 vs. 39%, 63/160, *p* = 0.14). SARS-CoV-2 anti-spike median IgG levels were not significantly different between these two groups (2.7 vs. 5.5 BAU/mL, *p* = 0.42, Figure 1).

In contrast, when considering only the 12 patients with longitudinal follow-up after both the second and third dose, an increase in anti-spike IgG levels was observed (0.19 vs. 5.28 BAU/mL, *p* = 0.03, Figure 1). In this subgroup, prevalence of positive SARS-CoV-2 anti-spike IgG and anti-spike IgG ≥ 30 BAU/mL did not increase after the third dose (17%, 2/12 vs. 42%, 5/12, *p* = 0.37 and 17%, 2/12 vs. 33%, 4/12, *p* = 0.64).

Age was associated with poor response to COVID-19 vaccines (>65 year, *p* = 0.02) but a history of COVID-19 RT-PCR was associated with better responses (*p* = 0.04, Table 1). Nine patients had a history of COVID-19, which occurred at a median of 206 days (IQR: 168–248) before the measurement of SARS-CoV-2 anti-spike IgG.

The third dose of vaccine was injected at a median of 43 days (IQR: 33–63) after the second dose. SARS-CoV-2 anti-spike IgG was measured at a median of 95 days (IQR: 48–124) and 52 days (IQR: 34–76) after the second and third dose of vaccine, respectively.

### 3.2. Patients Treated for Chronic Lymphocytic Leukemia

Prevalence of positive SARS-CoV-2 anti-spike IgG and prevalence of SARS-CoV-2 anti-spike IgG ≥ 30 BAU/mL in 51 patients treated for CLL did not increase between the second and third dose of vaccine (57%, 29/51 vs. 50%, 10/20, *p* = 0.61 and 45%, 23/51 vs. 45%, 9/20, *p* = 1.0, Table 1). SARS-CoV-2 anti-spike median IgG levels were comparable between these two groups (12.9 vs. 10.7 BAU/mL, *p* = 0.32, Figure 1).

In contrast, when considering only the 20 patients with longitudinal follow-up after both the second and third dose, we observed an increase in anti-spike IgG levels (0.63 vs. 10.7 BAU/mL, *p* = 0.0002, Figure 1) and an increase in the prevalence of anti-spike IgG ≥ 30 BAU/mL (5%, 1/20 vs. 45%, 9/20, *p* = 0.008). In this subgroup, prevalence of positive SARS-CoV-2 anti-spike IgG did not increase after the third dose (20%, 4/20 vs. 50%, 10/20, *p* = 0.10).

Treatment with tyrosine kinase inhibitors or venetoclax was associated with poor response to the second dose of COVID-19 vaccine (*p* = 0.007, Table 1). The third dose of vaccine was injected at a median of 63 days (IQR: 48–81) after the second dose. SARS-CoV-2 anti-spike IgG was measured at a median of 43 days (IQR: 36–57) and 42 days (IQR: 31–45) after the second and third dose of vaccine, respectively.

## 4. Discussion

In 12 kidney transplant recipients with longitudinal follow-up, a moderate increase in SARS-CoV-2 anti-spike IgG levels was observed after the third dose of COVID-19 vaccine (0.19 vs. 5.28 BAU/mL, *p* = 0.03). This increase in IgG levels had no significant impact on the prevalence of anti-spike IgG ≥ 30 BAU/mL (17%, 2/12 vs. 33%, 4/12, *p* = 0.64). In 20 patients treated for CLL with longitudinal follow-up, a moderate increase in SARS-CoV-2 anti-spike IgG levels was observed after the third dose of COVID-19 vaccine (0.63 vs. 10.7 BAU/mL, *p* = 0.0002). This resulted in an increase in the prevalence of anti-spike IgG ≥ 30 BAU/mL (5%, 1/20 vs. 45%, 9/20, *p* = 0.008). When considering all patients, with or without longitudinal follow-up, prevalence of anti-spike IgG ≥ 30 BAU/mL after the third dose was 39% (63/160) in kidney transplant recipients and 50% (10/20) in patients treated for CLL. Overall, this suggests that a third dose of COVID-19 vaccine in immunocompromised patients can increase antibody levels. Still, this increase might not be clinically relevant because few patients reached anti-spike IgG levels ≥ 30 BAU/mL, previously associated with 50% vaccine effectiveness [19]. These results are in line with other studies in kidney transplant recipients [10,11,12] and provide new data regarding patients with CLL.

In kidney transplant recipients, our results confirm the association between young age (<65 years) and response to COVID-19 vaccines (*p* = 0.02) [6,20] and suggest a better response in patients with a history of COVID-19 (*p* = 0.04). In patients treated for CLL, our results confirm the association between treatment with tyrosine kinase inhibitors or venetoclax and non-response to COVID-19 vaccines (*p* = 0.007) [8].

One limitation of our study is the small number of patients with longitudinal follow-up. Indeed, antibody responses in immunocompromised patients are probably too heterogeneous to compare groups of unpaired patients. Other limitations of our study are the lack of antibody measurements prior to vaccination, the lack of in vitro investigation of neutralizing Ab titers, especially against circulating variants, and cellular responses. Other parameters such as disease stage and other treatments, not included in this study, could also have an impact on response to COVID-19 vaccines.

Overall, we believe these results highlight the need for vaccination of the general population and the importance of upholding non-medical preventive measures to protect immunocompromised patients, especially those at higher risk of non-response to COVID-19 vaccines.

## Figures and Tables

**Figure 1 vaccines-09-01055-f001:**
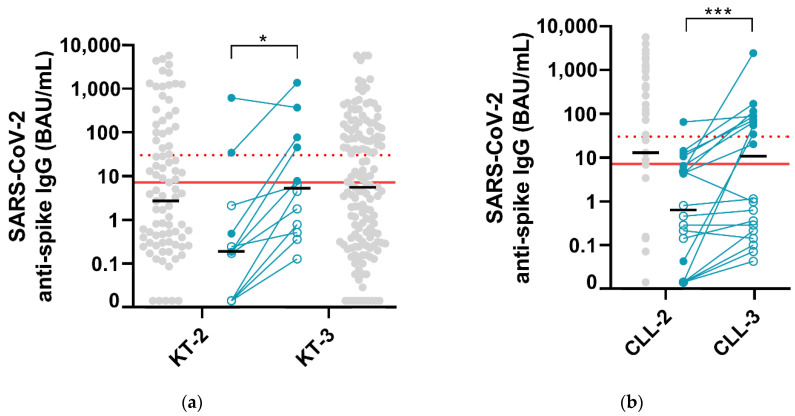
(**a**) SARS-CoV-2 anti-spike IgG levels measured in 245 kidney transplant (KT) recipients after the second (KT-2, n = 97) and third dose of COVID-19 vaccine (KT-3, n = 160). (**b**) SARS-CoV-2 anti-spike IgG levels measured in patients treated for chronic lymphocytic leukemia (CLL) after the second (CLL-2, n = 51) and the third dose of vaccine (CLL-3, n = 20). Longitudinal follow-up was performed for 12 KT patients and 20 CLL patients (responders and non-responders to the 3rd dose of vaccine in plain and empty blue dots, respectively). Light grey dots, patients without longitudinal follow-up; plain red bar, SARS-CoV-2 IgG II Quant assay cut-off (≥7.1 BAU/mL); dashed red bar, anti-spike IgG levels previously associated with 50 % vaccine effectiveness against symptomatic COVID-19 (≥30 BAU/mL) [19]; black bars, medians. * *p* < 0.05; *** *p* < 0.001.

**Table 1 vaccines-09-01055-t001:** Factors associated with response to COVID-19 vaccines in 245 kidney transplant recipients and 51 patients treated for chronic lymphocytic leukemia.

	Kidney Transplant						Chronic Lymphocytic Leukemia					
	After 2nd Dose (n = 97)			After 3rd Dose (n = 160)			After 2nd Dose (n = 51)			After 3rd Dose (n = 20)		
**Response to COVID vaccine**	No (n = 55)	Yes (n = 42)	*p*	No (n = 85)	Yes (n = 75)	*p*	No (n = 22)	Yes (n = 29)	*p*	No (n = 10)	Yes (n = 10)	*p*
Age > 65 year	31 (56)	13 (31)	**0.02**	41 (48)	26 (35)	0.08	17 (77)	20 (69)	0.55	8 (80)	6 (60)	0.63
Age < 50 year	5 (9)	8 (19)	0.23	15 (18)	24 (32)	0.06	0	1 (4)	1.0	0	0	-
Sex (female)	18 (33)	21 (5)	0.10	36 (42)	21 (28)	0.07	7 (32)	11 (38)	0.77	2 (20)	4 (40)	0.64
History of positive SARS-CoV-2 RT-PCR	1 (1.8)	6 (14)	**0.04**	0	2 (2.7)	0.22	-	-	-	-	-	-
COVID-19 vaccine												
Pfizer (BNT162b2)	43 (78)	32 (76)	0.81	53 (62)	45 (60)	0.88	-	-	-	-	-	-
Moderna (mRNA-1273)	3 (5.5)	4 (9.5)	0.46	11 (13)	16 (21)	0.20	-	-	-	-	-	-
Tyrosine kinase inhibitors or venetoclax	-	-	-	-	-	-	19 (86)	14 (48)	**0.007**	9 (90)	9 (90)	1.0
Prior-CLL directed therapy	-	-	-	-	-	-	15 (68)	12 (41)	0.54	9 (90)	7 (70)	0.58
Anti-CD20 monoclonal antibody within 2 years	-	-	-	-	-	-	2 (9)	0	0.18	1 (10)	0	1.0

Results are represented as median (range) or absolute values (percentages); *p* < 0.05 are in bold.

## Data Availability

Data supporting reported results can be provided by contacting the corresponding author.

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
