# Peer review of "Antibody Responses after a Third Dose of COVID-19 Vaccine in Kidney Transplant Recipients and Patients Treated for Chronic Lymphocytic Leukemia"

_vaccines, 2021, doi:10.3390/vaccines9101055_

Round 1
Reviewer 1 Report
In this manuscript, the authors check antibody response agains SARS-COV2 vaccines after 2nd and 3rd dose in two populations: kidney transplant and CLL.While one of the main weaknesses of this research is that data derived form cellular response are lacking (which is a better surrogate marker for response to vaccines) the results are interesting and should be shown. The conclusions are controversial, but with the data provided we can recommend immunocompromissed hosts should receive a third SARS-COV2 dose the earlier the better.
Reviewer 2 Report
Dr Julien Marlet and Coll wished to characterize the antibody responses induced by a third dose of mRNA COVID-19 vaccine in kidney transplant recipients and in patients treated for chronic lymphocytic leukemia in a retrospective study .
By defining responders those patients who showed positive SARS-CoV-2 IgG, corresponding to levels ≥ 7.1 binding antibody units or above the cut-off of 30 BAU/mL (binding antibodies unit) after the third dose, Authors found that only 47 % and 39% (among kidney transplant recipients and 57 % and 50 % in patients treated for CLL could be classified as responders.
In Discussion Authors comment that, even if third dose of COVID-19 vaccine in immune-compromised patients can increase antibody levels, the observed increase might not be clinically relevant because few patients reached anti-spike IgG levels ≥ 30 BAU/mL, previously associated with 50 % vaccine effectiveness.
We agree with the Authors about the importance of non-medical preventive measures to protect immunocompromised patients.
Minor points
1- The timing of antibodies measurements should be reported in Methods
2-Authors should add that an important limitation of their study is the lack of in vitro investigation of neutralizing Ab titers towards recently reported variants of Sars-Cov-2
Reviewer 3 Report
The paper raises the interesting issue of the response of immunocompromised patients to the anti COVID vaccine. Unfortunately, the paper has significant shortcomings:
1. extremely small group of subjects
2. inconsistent group of patients - different in terms of therapy, different in terms of type of vaccines, different in terms of COVID history, etc.
3. no determination of antibody levels prior to vaccination - no baseline
4. no data on time interval from COVID to vaccination
5. For CLL patients, no data on disease stage, treatment, CD20, CD3, CD4, CD8 cell count, etc.
6. in the case of kidney transplant patients, no data on what kind of immunosuppressive drugs they receive, also no data described above for CLL
The authors conclude that the cocoon theory of vaccination is also relevant to the anti Covid vaccine.
Round 2
Reviewer 3 Report
I thank the authors for making the changes. The paper has its limitations as described in the discussion section, but nevertheless the topic of the paper is valuable.